# The Peroxisome Proliferator-Activated Receptor α- Agonist Gemfibrozil Promotes Defense Against *Mycobacterium abscessus* Infections

**DOI:** 10.3390/cells9030648

**Published:** 2020-03-06

**Authors:** Yi Sak Kim, Jin Kyung Kim, Bui Thi Bich Hanh, Soo Yeon Kim, Hyeon Ji Kim, Young Jae Kim, Sang Min Jeon, Cho Rong Park, Goo Taeg Oh, June-Woo Park, Jin-Man Kim, Jichan Jang, Eun-Kyeong Jo

**Affiliations:** 1Department of Microbiology, Chungnam National University School of Medicine, Daejeon 35015, Korea; isaackim@cnu.ac.kr (Y.S.K.); pcjlovesh6@naver.com (J.K.K.); hjee0226@hanmail.net (H.J.K.); siestera@hanmail.net (Y.J.K.); valersangmin@gmail.com (S.M.J.); dd910817@gmail.com (C.R.P.); 2Infection Control Convergence Research Center, Chungnam National University School of Medicine, Daejeon 35015, Korea; 3Molecular Mechanisms of Antibiotics, Division of Life Science, Research Institute of Life Science, Gyeongsang National University, Jinju 52828, Korea; hanhm0515006@gstudent.ctu.edu.vn; 4Division of Applied Life Science (BK21plus Program), Gyeongsang National University, Jinju 52828, Korea; 5Drug & Disease Target Research Team, Division of Bioconvergence Analysis, Korea Basic Science Institute (KBSI), Cheongju 28119, Korea; ksy1212@kbsi.re.kr; 6Department of Life Science, Ewha Womans University, Seoul 03760, Korea; gootaeg@ewha.ac.kr; 7Environmental Risk Assessment Research Division, Korea Institute of Toxicology, Jinju 52834, Korea; jwpark@kitox.re.kr; 8Human and Environmental Toxicology Program, Korea University of Science and Technology (UST), Daejeon 34113, Korea; 9Department of Pathology, Chungnam National University School of Medicine, Daejeon 35015, Korea; jinmank@cnu.ac.kr

**Keywords:** PPARα, *Mycobacterium abscessus*, gemfibrozil, TFEB, inflammation

## Abstract

Peroxisome proliferator-activated receptor α (PPARα) shows promising potential to enhance host defenses against *Mycobacterium tuberculosis* infection. Herein we evaluated the protective effect of PPARα against nontuberculous mycobacterial (NTM) infections. Using a rapidly growing NTM species, *Mycobacterium abscessus* (Mabc), we found that the intracellular bacterial load and histopathological damage were increased in PPARα-null mice in vivo. In addition, PPARα deficiency led to excessive production of proinflammatory cytokines and chemokines after infection of the lung and macrophages. Notably, administration of gemfibrozil (GEM), a PPARα activator, significantly reduced the in vivo Mabc load and inflammatory response in mice. Transcription factor EB was required for the antimicrobial response against Mabc infection. Collectively, these results suggest that manipulation of PPARα activation has promising potential as a therapeutic strategy for NTM disease.

## 1. Introduction

The incidence of nontuberculous mycobacterial (NTM) infection is increasing worldwide, not only in immunocompromised patients, but also in healthy individuals [1,2,3,4]. However, little is known about the host factors that influence the pathogenesis of NTM infection [1,3,4]. *Mycobacterium abscessus* (Mabc), a rapidly growing NTM species, lives in natural environments such as water and soil, and causes pulmonary, cutaneous, and other opportunistic infectious diseases [3,5]. Although the pulmonary infection caused by Mabc progresses slowly, it is often virulent and difficult to treat with antimicrobials due to its resistance to most antibiotic classes [5]. A better understanding of host defense mechanisms is needed to develop novel control strategies against infection with Mabc and other NTM.

Peroxisome proliferator-activated receptor α (PPARα) is an essential nuclear receptor that plays a critical role in lipid metabolism (i.e., fatty acid β-oxidation) and regulation of inflammation in various cell types and tissues [6,7,8]. Due to its regulation of macrophage-mediated inflammation, PPARα is a promising target for various human pathologies, particularly cardiovascular and metabolic diseases [9,10]. In addition, PPARα expression is involved in protective role against numerous disease models including acute lung injury [11], sepsis-associated acute kidney injury [12], non-alcoholic steatohepatitis [13], etc. Indeed, PPARα as well as PPARγ play crucial roles in the modulation of host defenses against a variety of microbial infections [14,15,16,17] . In particular, PPARα functions in controlling the replication of *Pseudomonas aeruginosa* [18] and in the activation of antiviral metabolism during hepatitis C virus infections [19]. Furthermore, PPARα promotes defense against *Mycobacterium tuberculosis* infection in macrophages by regulating excessive inflammation, inducing the expression of transcription factor EB (TFEB), and upregulating lipid catabolism [20]. However, the role of PPARα in innate host defense against Mabc infection remains to be characterized. In addition, it is urgently needed to develop new therapeutics against NTM disease, in particular, Mabc infection, which is often considered the cause of serious pulmonary infections and unfortunately associated with a high rate of treatment failure [21].

In this study, we evaluated whether PPARα is essential for Mabc infection in vivo and in vitro. PPARα deficiency led to an increased bacterial load and excessive inflammatory response during Mabc infection in vivo and in vitro. In addition, PPARα-null macrophages exhibited reduced nuclear translocation of TFEB and phagosomal maturation during Mabc infection, compared with wild-type cells. Furthermore, the PPARα agonist gemfibrozil (GEM) exerted a therapeutic effect and an anti-inflammatory response against Mabc infection in mice.

## 2. Materials and Methods

### 2.1. Mice

In this study, 8–12-week-old (age-matched) male C57BL/6 (Samtako Bio, Gyeonggi-do, Korea), *Ppara*^+/+^ (littermate control, inbred), and *Ppara*^−/−^ mice on a C57BL/6 background (B6.129S4-Ppara^tm1Gonz^/J) were used. *Ppara*^−/−^ mice were kindly provided by Dr. Gonzalez [22]. Mice were maintained on a 12 h light/dark cycle under specific pathogen-free conditions. All animal experimental methods and procedures were performed in accordance with the relevant ethical guidelines and regulations. This study was approved by the Institutional Research and Ethics Committee at Chungnam National University, School of Medicine (CNU-00944; Daejeon, Korea) and the Korea Research Institute of Bioscience and Biotechnology (project license; KCDC-15-3-01). All animal procedures were conducted in accordance with the guidelines of the Korean Food and Drug Administration.

### 2.2. Cells

Primary bone marrow-derived macrophages (BMDMs) were isolated from C57BL/6, *Ppara*^+/+^, and *Ppara*^−/−^ mice and cultured in Dulbecco’s modified Eagle’s medium (Lonza, Basel, Swiss) containing 10% fetal bovine serum (Lonza), penicillin (100 IU/mL), and streptomycin (100 μg/mL). BMDMs were differentiated for 3–5 days in the presence of 25 ng/mL macrophage colony-stimulating factor (M-CSF; R&D Systems, Minneapolis, MN, USA), incubating in a 37 °C humidified atmosphere containing 5% CO_2_. Human PBMCs were isolated from heparinized venous blood using Ficoll-Hypaque (Lymphoprep; Axis-Shield, Dundee, Scotland, 1114545) as described previously [23]. For macrophage differentiation, adherent monocytes were incubated in Roswell Park Memorial Institute 1640 medium (Lonza, 12-702F) containing 10% pooled human serum (Lonza, 14-402), 1% L-glutamine, 50 IU/mL penicillin, and 50 mg/mL streptomycin for 1 h at 37 °C, after which the nonadherent cells were removed. Human monocyte-derived macrophages (MDMs) were prepared by culturing peripheral blood monocytes for 4 d in the presence of 4 ng/mL human CSF/macrophage colony-stimulating factor (Sigma-Aldrich, St. Louis, MO, USA, M6518) as described previously [24]. All experimental procedures were approved by the Chungnam National University Institutional Research and Ethics Committee.

### 2.3. Mycobacterial Strains and Culture

Mabc (ATCC 19977), Mabc (CIP 104536T), Mabc-enhanced green fluorescent protein (EGFP), and Mabc-LuxG13 were incubated at 37 °C with shaking in Middlebrook 7H9 broth (BD Biosciences, Franklin Lakes, NJ, USA, 271310) supplemented with 0.2% glycerol, 0.05% Tween-80 (Sigma-Aldrich), and albumin–dextrose catalase. Bacterial cultures were harvested by centrifugation at 500 × g for 20 min, and the pellets were resuspended in bacterial culture medium. Bacterial cultures were divided into 1 mL aliquots and stored at −80 °C. Mabc cultured cells in the logarithmic phase were centrifuged, washed, and resuspended in PBS plus 0.05% Tween 80 (PBST) before homogenizing through a 26-gauge needle and sonicated in a water bath. Bacterial stock cultures were stored at −80 °C. The number of colony-forming units (CFUs) of the inoculum was verified by plating serial dilutions. We used 50 μg/mL kanamycin to culture Mabc-EGFP. pMV306hsp+LuxG13 (plasmid #26161) was introduced into competent Mabc cells by electroporation and plated on 7H10 agar containing 50 μg/mL kanamycin. Kanamycin-resistant colonies were picked and inoculated into 7H9 broth containing kanamycin. Luminescent Mabc was identified using a SpectraMax M3 luminescent plate reader (Molecular Devices, San Jose, CA, USA). Bacteria in mid-logarithmic phase (absorbance 0.4) were used in all experiments. Representative vials were thawed and the CFU enumerated by serially diluting and plating on Middlebrook 7H10 agar (BD Biosciences, 262710).

### 2.4. Mycobacterial Infection

BMDMs were infected for 2 h with Mabc, Mabc-EGFP, or Mabc-LuxG13, washed three times with PBS to remove extracellular bacteria, and incubated in fresh medium for the indicated periods at 37 °C. For mouse infection in vivo, frozen Mabc cells were thawed and centrifuged, and the pellet was resuspended in PBST. Mice were administered with Mabc either intravenously (iv; 1 × 10^7^ CFU/mouse) or intranasally (in; 1 × 10^6^ CFU/mouse). To assess the bacterial burden, mice were euthanized at the indicated times after Mabc infection, and the lung, spleen, and liver were removed and homogenized using a tissue homogenizer (OMNI TH) in PBST. To assess bacterial viability in BMDMs, infected cells were lysed in distilled water to release intracellular bacteria. Thereafter, the serially diluted homogenates of Mabc-infected tissues and cells were inoculated onto 7H10 agar, and colonies were counted 3–4 days later. To quantify intracellular Mabc-LuxG13 based on luciferase activity, the luminescence signal of Mabc-LuxG13 was measured for 10 s using a luminescence microplate reader, and bioluminescence was expressed as relative light units [25].

### 2.5. Antibodies and Reagents

Anti-β-actin (sc-1616-R) antibodies were purchased from Santa Cruz Biotechnology (Dallas, TX, USA). Anti-TFEB (A303-673A) antibody was purchased from Bethyl Laboratories (Montgomery, TX, USA). Alexa Fluor 594-conjugated anti-rabbit IgG (A21207) was purchased from Invitrogen (Carlsbad, CA, USA). GW7647 (1008613) and GW6471 (880635-03-0) were obtained from Cayman Chemical (Ann Arbor, MI, USA). Anti-LC3A/B (L8918) and anti-p62 (P0067) antibodies, GEM (G9518), and clarithromycin (C9742) were purchased from Sigma-Aldrich.

### 2.6. GEM Treatment

GEM was dissolved in dimethyl sulfoxide (DMSO) followed by dilution in 0.1% methyl cellulose (Sigma-Aldrich, M7027) and was used at 15 mg/kg body weight. Mice were treated with vehicle (0.1% methyl cellulose) or GEM once daily via gavage from days 3 to 7 after infection.

### 2.7. Immunopathology of Mycobacterium-Infected Mice

Lungs were harvested from mice infected with Mabc for the indicated times, fixed in 10% formalin, and embedded in paraffin wax. For histopathology, lung paraffin sections (4 µm) were cut and stained with hematoxylin and eosin (H&E). H&E-stained images were scanned using the Aperio Digital Pathology Slide Scanner (Leica, Wetzlar, Germany) and imaged using the ScanScope® CS System (Leica). The histopathological severity score was graded by scanning multiple random fields in six sections per mouse. For immunohistochemical (IHC) staining, lung paraffin sections (4 µm) were cut and immunostained with antibodies specific for myeloperoxidase (MPO; Abcam, Cambridge, MA, USA, ab9535) and LyG6 (Bio X Cell, West Lebanon, NH, USA, BP0075-1). Immunohistochemically stained lung tissue slides were examined using a confocal laser scanning microscope.

### 2.8. RNA Extraction, Semiquantitative Real-Time PCR, and Quantitative Real-Time PCR

Total RNA was isolated from homogenized tissues or cells using TRIzol (Thermo Fisher Scientific, Waltham, MA, USA, 15596-026), following the manufacturer’s instructions. After RNA quantitation, cDNA was synthesized by reverse transcription using the reverse transcriptase premix (Elpis Biotech, Daejeon, Korea, EBT-1515). Semiquantitative PCR was performed using the Prime Taq Premix (GeNet Bio, Nonsan, Korea, G-3000). Quantitative real-time PCR (qRT-PCR) was performed on the Rotor-Gene Q 2plex system (Qiagen, Hilden, Germany) using SYBR Green master mix (Qiagen, 204074). Gene expression was relatively quantitated using the 2^−ΔΔCt^ method, with normalization to β-actin expression, and expressed as the relative fold change. The primer sequences used are listed in Appendix A.

### 2.9. Enzyme-Linked Immunosorbent Assay

TNF, IL-6, and IL-12p40 levels in cell supernatants were analyzed using mouse ELISA kits (BD Biosciences; TNF, 558534 and IL-6, 555240) according to the manufacturer’s instructions.

### 2.10. Immunoblotting

For immunoblotting, cells were lysed in RIPA (50 mM Tris-HCl at pH 7.5, 2 mM ethylenediaminetetraacetic acid, 150 mM NaCl, 0.1% sodium dodecyl sulfate [SDS], 1% sodium deoxycholate, and 1% Triton X-100) supplemented with protease inhibitor mixture (Roche, Basel, Swiss). The protein extracts were boiled in 1× SDS sample buffer, resolved by SDS–polyacrylamide gel electrophoresis, and transferred to polyvinylidene difluoride membranes (Millipore, Burlington, MA, USA, IPVH00010). The membranes were blocked in 5% nonfat milk in PBST (3.2 mM Na_2_HPO_4_, 0.5 mM KH_2_PO_4_, 1.3 mM KCl, 135 mM NaCl, and 0.05% Tween 20 [Sigma-Aldrich], pH 7.4) for 1 h and incubated with the appropriate antibodies. Protein signals were visualized using enhanced chemiluminescence solution (Millipore, WBKL S0500) and detected using the UVitec Alliance mini-chemiluminescence device (UVitec, Rugby, UK).

### 2.11. Lentiviral shRNA Transduction

For silencing of mouse *Tfeb* in primary cells, the three packaging plasmids pRSV-Rev [12253], pMD2.VSV-G [12259], and pMDLg/pRRE [12251] (Addgene, Watertown, MA, USA, deposited by Dr. Didier Trono) and mouse *Tfeb* shRNA (Santa Cruz Biotechnology, sc-38510-SH) were cotransfected into HEK293T cells using Lipofectamine 2000 (Invitrogen, 11668-019). After 72 h, virus-containing supernatant was collected and filtered. BMDMs were seeded into 24-well plates and incubated with lentiviral vectors in the presence of 4 μg/mL polybrene (Sigma-Aldrich, 107689) for 24 h. On the following day, the medium was replaced with fresh medium. After incubation for 3 days, the transduction efficiency was analyzed by PCR.

### 2.12. Immunofluorescence

After the appropriate treatment, cells on coverslips were washed three times with PBS, fixed with 4% paraformaldehyde for 15 min, permeabilized with 0.25% Triton X-100 (Sigma-Aldrich, St. Louis, MO, USA) for 10 min, and incubated with the appropriate primary antibody overnight at 4 °C. Cells were washed with PBS to remove excess primary antibody and incubated with the secondary antibody for 2 h at room temperature. Nuclei were stained with DAPI for 2 min. After mounting, fluorescence images were acquired and analyzed using the TCS SP8 confocal microscope (Leica, Wetzlar, Germany) and MetaMorph Advanced Imaging acquisition software version 7.8 (Molecular Devices, San Jose, CA, USA).

### 2.13. Minimum Inhibitory Concentration (MIC) Determination Using Resazurin Microtiter Assay (REMA)

The MICs of the compounds were determined as described previously [26]. Briefly, 100 μL of cation-adjusted Mueller–Hinton (CAMH) and 7H9 broth supplemented with 10% ADC was added to every well of a 96-well microtiter plate. A two-fold serial dilutions were prepared in 96-well clear microplates to obtain concentration ranges from 200 μM to 0.39 nM. Log-phase Mabc strain CIP 104536T cultures were added to each wells to an OD600 of 0.05 and the plates were incubated at 37 °C for 3 days prior to addition of resazurin (0.025% [wt/vol]). After overnight incubation, fluorescence was measured by excitation at 530 nm and emission at 590 nm using a SpectraMax® M3 Multi-Mode Microplate Reader (Molecular Devices, San Jose, CA, USA). The MIC was calculated using GraphPad Prism 6 software (GraphPad Software, Inc., La Jolla, CA, USA).

### 2.14. Statistical Analysis

Each experiments was carried out three times. Statistical tests were performed using GraphPad Prism version 5 by Student’s t-test and one-way ANOVA. Data are presented as means ± standard deviation (SD). 

## 3. Results

### 3.1. Ppara^-/-^ Mice Exhibited an Upregulated Proinflammatory but Defective Antimicrobial Response During Mabc Infection Compared with Ppara^+/+^ Mice

To examine whether PPARα is involved in the defense against Mabc infection, we infected *Ppara*^+/+^ and *Ppara*^-/-^ mice with Mabc (ATCC 19977; iv, 1 × 10^7^ CFU/mouse) for 7 days and examined the mycobacterial burden in the lung, liver, and spleen. As shown in Figure 1A, the in vivo bacterial load was significantly increased in the lung, liver, and spleen of *Ppara*^-/-^ mice compared with *Ppara*^+/+^ mice at 7 days post-infection (dpi). To confirm that PPARα is required for host defense, we infected mice with Mabc (in, 1 × 10^6^ CFU/mouse). The number of bacteria was greater in the lung of *Ppara*^-/-^ mice than *Ppara*^+/+^ mice (Figure 1B).

We next examined the histopathological features on the lungs of *Ppara*^+/+^ and *Ppara*^-/-^ mice infected with Mabc. As shown in Figure 1C, the number of granuloma-like inflammatory lesions was significantly increased in *Ppara*^-/-^ mice compared with *Ppara*^+/+^ mice. Neutrophil infiltration in the lungs may play a protective or pathological role during Mabc infection [27,28]. Thus, we examined neutrophil infiltration in the lungs of *Ppara*^+/+^ and *Ppara*^-/-^ mice. 

Notably, the LyG6 (Figure 1D) and MPO (Appendix A) staining intensity was increased in the lung of *Ppara*^-/-^ mice infected with Mabc, suggesting that the neutrophil-mediated inflammatory response is markedly increased in the lung of *Ppara*^-/-^ compared with *Ppara*^+/+^ mice. Furthermore, intracellular survival of Mabc was significantly increased in BMDMs from *Ppara*^-/-^ mice compared with *Ppara*^+/+^ mice (Figure 1E). These results indicate that PPARα is essential for the antimicrobial response and for controlling neutrophil infiltration during Mabc infection.

### 3.2. PPARα Deficiency Increased the Inflammatory Response During Mabc Infection

PPARα is a regulator of the inflammatory response. We next examined whether PPARα is involved in Mabc-induced inflammation in vivo. The mRNA levels of proinflammatory cytokines and chemokines were markedly upregulated in the lung of *Ppara*^-/-^ compared with *Ppara*^+/+^ mice (Figure 2A). Notably, the mRNA levels of *Tnf*, *Il6*, *Il1b*, *Cxcl10*, and *Cxcl5* were significantly increased and that of *Il10* decreased in lung tissue from *Ppara*^-/-^ mice, compared with *Ppara*^+/+^ mice, during Mabc infection (Figure 2A).

We next examined the mRNA and protein levels of these cytokines and chemokines in *Ppara*^+/+^ and *Ppara*^-/-^ BMDMs after Mabc infection. As shown in Figure 2B, the mRNA levels of *Tnf*, *Il6*, and *Il1b* were significantly increased in BMDMs from *Ppara*^-/-^ mice, compared with *Ppara*^+/+^ mice, at 6 h after Mabc infection. In addition, the TNF and IL-6 protein levels were markedly upregulated in *Ppara*^-/-^ BMDMs compared with *Ppara*^+/+^ BMDMs, at 6 and 18 h after Mabc infection (Figure 2C). Furthermore, the expression of TNF, IL-6, and IL-1β mRNA and protein levels were significantly increased in *Ppara*^-/-^ BMDMs, compared with *Ppara*^+/+^ BMDMs, after Mabc infection (Figure 2D,E; protein and mRNA, respectively). Together, these data suggest that proinflammatory responses are markedly upregulated in the lungs and macrophages from *Ppara*^-/-^ mice, compared with *Ppara*^+/+^ mice, during Mabc infection.

### 3.3. PPARα Activation Enhances Macrophage Antimicrobial Responses Againt Mabc Infection

We next investigated the effects of PPARα agonists on Mabc infection in macrophages. Although PPARα agonist GEM [29] did not exhibit any direct antimicrobial effect against Mabc (Appendix A), it substantially inhibited the intracellular growth of Mabc in BMDMs (Figure 3A,B) and human MDMs (Figure 3C). Notably, GEM (100 μM) treatment of BMDMs and human MDMs resulted in significant inhibition of the intracellular growth of Mabc (Figure 3A–C). As a positive control, treatment of Mabc-infected BMDMs with clarithromycin resulted in a marked inhibitory effect on intracellular growth of Mabc (Figure 3B). However, it is noted that clarithromycin is a drug with pleiotropic activity [30]. In addition, the combined treatment of BMDMs with GEM and clarithromycin did not show any significant additional/synergistic effect upon anti-Mabc response (data not shown). Moreover, GEM-mediated antimicrobial activity was mediated through PPARα, since the GEM-induced suppressive effect on intracellular growth of Mabc in *Ppara*^+/+^ BMDMs was significantly abrogated in *Ppara*^-/-^ BMDMs (Figure 3D). We also showed that application of the PPARα antagonist GW6471 [31] increased intracellular bacterial growth in a dose-dependent manner (Figure 3A). These data suggest that GEM treatment increased the antimicrobial activity against intracellular Mabc through PPARα.

### 3.4. PPARα Activation Regulates the Inflammatory Response to Mabc Infection

We next examined regulation of the inflammatory response by GEM during Mabc infection. As shown in Figure 4A, GEM treatment (100 μM) of BMDMs dramatically suppressed the expression of a variety of inflammatory cytokines and chemokines (*Il6*, *Il12p40*, *Il1b*, *Cxcl10*, and *Cxcl5*) at 18 h after infection. The *Il10* mRNA level was substantially decreased in BMDMs at 18 h but significantly increased at 6 h after Mabc infection (Figure 4B). In addition, we examined whether GEM treatment inhibited Mabc-induced inflammatory responses in macrophages through PPARα. To examine this, we infected *Ppara*^+/+^ and *Ppara*^-/-^ BMDMs with Mabc (MOI of 5), and treated with GEM (100 μM) for 18 h prior to measure the mRNA levels of *Il6* and *Il1b*. As shown in Figure 4C, we found that the inhibitory effects of GEM on Mabc-mediated *Il6* and *Ilb* expression in *Ppara*^+/+^ BMDMs were significantly abrogated in *Ppara*^-/-^ BMDMs. Therefore, GEM treatment suppressed Mabc-induced inflammatory response depending on PPARα.

### 3.5. PPARα Activation Upregulates Antimicrobial Responses and Downregulates Pathological Inflammation during Mabc Infection

To confirm and extend the in vitro data, we next assessed the function of PPARα activation by GEM in the defense against Mabc infection in vivo. C57BL/6 mice were infected with Mabc iv, and GEM (15 g/kg body weight) was administered orally. As shown in Figure 5A,B, the numbers of viable bacteria in the lung, liver, and spleen were significantly reduced in GEM-treated mice compared with control mice (Figure 5A,B, iv and in, respectively). Consistent with the in vitro results (Figure 4A), the mRNA levels of proinflammatory cytokines (*Tnf* and *Il1b*) were significantly decreased in the lung of GEM-treated mice compared with control mice (Figure 5C). In addition, the *Il10* mRNA level was significantly upregulated in the lung tissue of GEM-treated mice compared with control mice (Figure 5C). 

A rough variant of Mabc enhanced type I interferon (IFN) production and cell death [32]. We thus examined the expression of *Ifna5* in the lung tissues of *Ppara*^+/+^ compared with *Ppara*^-/-^ mice. Interestingly, the *Ifna5* mRNA level was markedly upregulated in the lung of Mabc-infected *Ppara*^-/-^ compared with *Ppara*^+/+^ mice (Figure 5C). Moreover, the number of granulomatous lesions was considerably decreased by GEM (Figure 5D).

### 3.6. PPARα is Required for Nuclear Translocation of TFEB in Macrophages during Mabc Infection

The smooth variant of Mabc restricts phagosomal acidification and induces less autophagy compared with the rough variant [33]. We thus evaluated LC3-I/II conversion and p62 expression in BMDMs after Mabc infection. As shown in Figure 6A, Mabc infection increased the LC3-II and p62 levels, suggesting blocking of autophagic flux. In addition, phagosomal acidification, *i.e.*, colocalization of Mabc with the lysosomal marker LAMP1, was markedly reduced in *Ppara*^-/-^ BMDMs compared with *Ppara*^+/+^ at 4 h (Figure 6B,C). At 24 h after Mabc infection, the slight intraphagosomal acidification disappeared in *Ppara*^+/+^ BMDMs such that there was no difference between *Ppara*^+/+^ and *Ppara*^-/-^ BMDMs (Figure 6B,C).

TFEB is an essential transcription factor for genes involved in autophagy and lysosomal biogenesis [34]. TFEB activation is required for IFNγ- or PPARα-induced autophagy and restriction of the intracellular growth of *M. tuberculosis* [20,35]. However, the nuclear translocation of TFEB has not been investigated in the context of Mabc infection. We compared the intracellular translocation of TFEB in BMDMs after Mabc infection between *Ppara*^+/+^ and *Ppara*^-/-^ BMDMs. As shown in Figure 6D, we found that TFEB nuclear translocation was significantly increased in *Ppara*^+/+^ BMDMs, but markedly decreased in *Ppara*^-/-^ BMDMs, after 30 min of Mabc infection. 

We next examined whether PPARα activation by GEM induces nuclear translocation of TFEB in BMDMs. GEM treatment of BMDMs led to a significant increase in TFEB nuclear translocation in Mabc-infected BMDMs (Figure 6E). We further examined whether other PPARα agonist regulated nuclear translocation of TFEB. As shown in Appendix A, treatment of BMDMs with GW7647 did not significantly upregulate the nuclear translocation of TFEB in BMDMs during Mabc infection, when compared with untreated controls. Furthermore, the GEM-mediated antimicrobial response in BMDMs was significantly attenuated by knockdown of *Tfeb* using shRNA targeting *Tfeb*, compared with a nonspecific shRNA (Figure 6F). Together, these data suggest that PPARα activation by GEM upregulates phagosomal maturation and the antimicrobial response to Mabc by inducing nuclear translocation of TFEB.

## 4. Discussion

Mabc, a rapidly growing NTM, causes serious pulmonary infections in both immunocompetent and immunocompromised individuals [1,36,37]. Within the Mabc complex, Mabc (*sensu stricto*) is a clinically important species because it is the most drug-resistant NTM and is associated with a high rate of treatment failure [37,38]. Understanding host–pathogen interactions and the underlying mechanisms is vital for the development of new therapeutics against NTM infection [1,39,40]. Recent studies on the roles of the innate and acquired immune system in NTM infection have suggested that multiple pathways/factors contribute to the pathogenesis of Mabc infection and host protective immunity [1,40]. In this study, we found that activation of PPARα induced an antimicrobial response to Mabc infection in vitro and in vivo.

PPARα is a key factor involved in host defense in macrophages against *M. tuberculosis* and *M. bovis* BCG [20]. These findings raised the question of whether PPARα promotes innate host defense against NTM infections and, therefore, whether PPARα-activating agents have potential as drug targets for Mabc infection in vivo. We addressed this issue using in vivo and in vitro models of Mabc infection and found that BMDMs from *Ppara*^-/-^ mice exhibited increased bacterial growth compared with those from *Ppara*^+/+^ mice. Similar to the findings of previous studies of *M. tuberculosis* and *M. bovis* BCG [20], we found that PPARα deficiency increased the levels of proinflammatory cytokines and chemokines in the lung and macrophages during Mabc infection. These findings are in partial agreement with a recent report of a protective role of PPARα activation against lipopolysaccharide-induced acute lung injury in mice [11].

In this study, we found that neutrophil infiltration was significantly increased in *Ppara*^-/-^ BMDMs and mice after infection. Therefore, the inflammatory response presumably mediated by neutrophil infiltration may play a double-edged role in Mabc infection. Neutrophils have crucial functions in host defense, *i.e.*, bacterial uptake and mycobactericidal activity, against the rough and smooth morphotypes by generating reactive oxygen species and extracellular traps [41]. In addition, impaired TNF signaling and defective neutrophil trafficking result in formation of aberrant granulomas, massive bacterial growth, and death of zebrafish larvae [42]. In line with the clinical finding that anti-TNF therapy exacerbates Mabc infection [43,44], appropriate induction of TNF and other inflammatory cytokines as well as neutrophil infiltration may play a fundamental role in host defense against Mabc infection. However, in human neutrophils, Mabc induces the expression of genes encoding chemokines and proinflammatory cytokines and shows augmented biofilm formation, which promote survival of the bacteria by exploiting the neutrophil-rich environment [45]. During pulmonary NTM infection, recruited neutrophils release elastase and metalloproteinase, damaging the airway epithelium and resulting in formation of microabscesses [40]. Therefore, an excessive inflammatory response due to infiltrated neutrophils may induce an exaggerated immunopathological reaction in vivo.

Our results reveal that *Cxcl10* and *Cxcl5* mRNA levels were significantly modulated by PPARα during Mabc infection in vivo. In the lung of *Ppara*^-/-^ mice, *Cxcl5* and *Cxcl10* expression levels were significantly increased. The chemokine CXCL5 is involved in the regulation of neutrophil influx and pulmonary neutrophil inflammation during *M. tuberculosis* infection [46,47,48]. In addition, the *Cxcl5* mRNA level was significantly increased in macrophages and epithelial cells in *Sirt3*-deficient lung during *M. tuberculosis* infection [47]. Neutrophil depletion by targeting Ly6G reportedly enhances host defense against *M. tuberculosis* infection in mice [46,47,48]. In addition, whether CXCL10 is increased in Mabc patients or Mabc infection models is unclear. M1 macrophages secrete proinflammatory cytokines and chemokines, including TNF-α, IL-6, IL-12, and CXCL10 [49]. The production of these cytokines and CXCL10 was markedly upregulated in the lung of *Ppara*^-/-^ compared with *Ppara*^+/+^ mice. The CXCL10 level was found to be increased in the plasma of untreated and treatment-refractory patients with *Mycobacterium avium* complex (MAC)-induced lung diseases, suggesting a role of CXCL10 as a marker of NTM disease [50]. Furthermore, the serum CXCL10 level was increased in patients with the fibrocavitary form of *M. avium* complex lung disease, compared with healthy controls, suggesting a role of CXCL10 in the pathogenesis of NTM diseases [51]. Together, our data suggest that CXCL10 and CXCL5 contribute to the pathogenesis of Mabc lung infection by recruiting and inducing over-activation of M1 macrophages and neutrophils, respectively. 

Having established a relationship between PPARα and TFEB, we next determined whether the PPARα–TFEB axis is required for host defense against Mabc infection. The S variant of Mabc was reported to induce autophagy [33], and we found that Mabc infection of wild-type BMDMs blocked autophagic flux and induced slight phagosomal acidification in *Ppara*^+/+^ BMDMs. Moreover, the nuclear translocation of TFEB was slightly, but significantly, increased in *Ppara*^+/+^ but not *Ppara*^-/-^ BMDMs. Because TFEB is a master regulator of, and essential transcription factor for, lysosomal biogenesis and control of xenophagy during *M. tuberculosis* infection [20,52], the regulatory mechanisms of TFEB activation are critical for the antimycobacterial response. Nuclear receptor subfamily 1, group D, member 1, an adopted orphan nuclear receptor, plays an important role in the regulation of TFEB transcription [53]. Although we did not examine the effect of TFEB on the inflammatory response during Mabc infection, we reported previously that TFEB silencing enhanced the inflammatory response of macrophages infected with *M. tuberculosis* [20]. Therefore, while the host autophagy response differs between *M. tuberculosis* and Mabc, the functional significance of PPARα–TFEB in the antimicrobial response is identical. 

We found that GEM was effective against Mabc infection in vivo using mouse model. A prior study reported the antimycobacterial activity of GEM in bacteriological medium and in macrophages [54]. Importantly, we found that GEM did not exert a direct antimycobacterial effect against Mabc. However, GEM treatment showed a significant antimicrobial effect against Mabc infection in vitro and in vivo, depending on PPARα. For in vivo treatment, GEM was administered to mice at a dose of 15 mg/kg of body weight/day via gavage, as previously described [55]. In adult humans, GEM is prescribed at a dose of 600–1200 mg/day, therefore dose used in our study (15 mg/kg) is human equivalent dose to mice [55]. Mechanistically, we found that GEM treatment specifically upregulated TFEB nuclear translocation in BMDMs during Mabc infection. Moreover, the GEM-induced antimicrobial response was significantly attenuated in TFEB-knockdown BMDMs. These data suggest the PPARα agonist GEM as a promising target for development of host-directed therapy against Mabc infection, since long term use of antibiotics may cause multidrug resistance in NTM infection by blocking lysosomal acidification and autophagic degradation [56]. Thus, GEM has two mechanisms of action: induction of an antimycobacterial effect and activation of the PPARα–TFEB axis. Further investigation of the effect of GEM on host defense against multidrug-resistant Mabc strains is needed.

## Figures and Tables

**Figure 1 cells-09-00648-f001:**
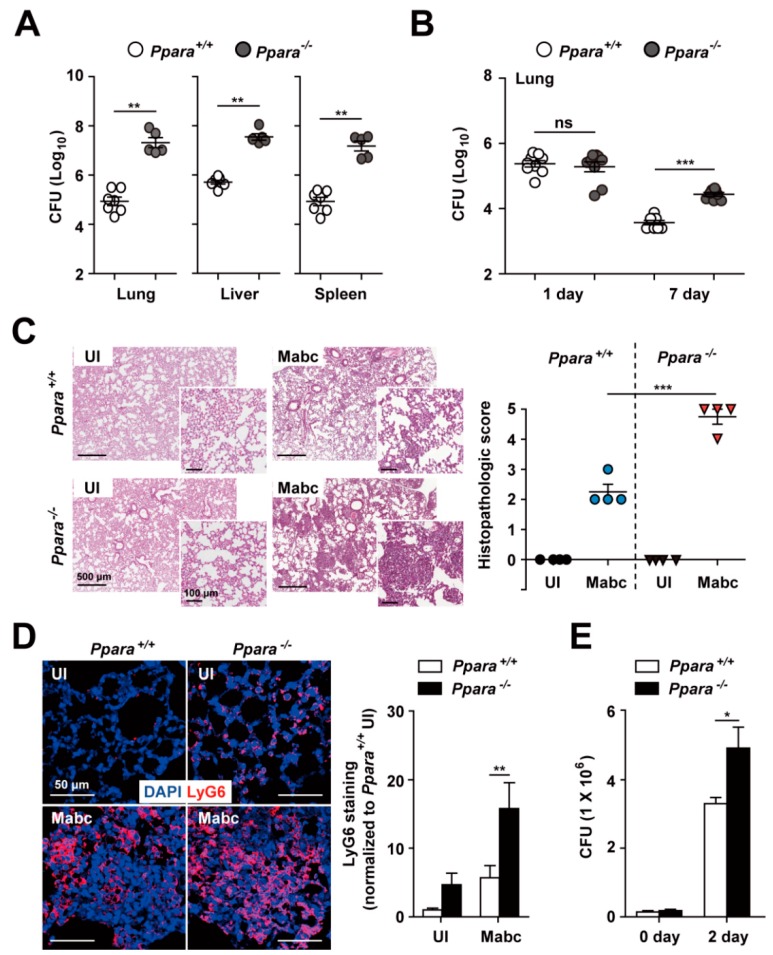
*Ppara*^-/-^ mice are more susceptible than *Ppara*^+/+^ mice to Mabc infection. (**A**) *Ppara*^+/+^ (n = 7) and *Ppara*^-/-^ (n = 5) mice were infected with Mabc (iv, 1 × 10^7^ CFU), and the bacterial loads in the lung, liver, and spleen were determined at 7 dpi. (**B**) *Ppara*^+/+^ (n = 9) and *Ppara*^-/-^ (n = 8) mice were infected with Mabc (in, 1 × 10^6^ CFU), and the bacterial load in the lung was determined at 7 dpi. (**C**,**D**) Lungs from Mabc-infected mice at 7 dpi as in (**A**). (**C**) (Left) Representative H&E-stained images. Scale bars, 500 and 100 μm. (Right) Quantitative analysis of histopathology scores. (**D**) (Left) Lung tissues were stained with LyG6 (red) and DAPI (nuclei; blue). Scale bars, 50 μm. (Right) Average mean fluorescence intensities (MFIs) of LyG6. (**E**) *Ppara*^+/+^ and *Ppara*^-/-^ BMDMs were infected with Mabc (MOI of 5) and intracellular survival of Mabc was determined at 2 dpi. Data are means ± SD of three independent experiments. Images are representative of three independent experiments (C and D). * *p* < 0.05, ** *p* < 0.01, and *** *p* < 0.001. ns, not significant; UI, uninfected.

**Figure 2 cells-09-00648-f002:**
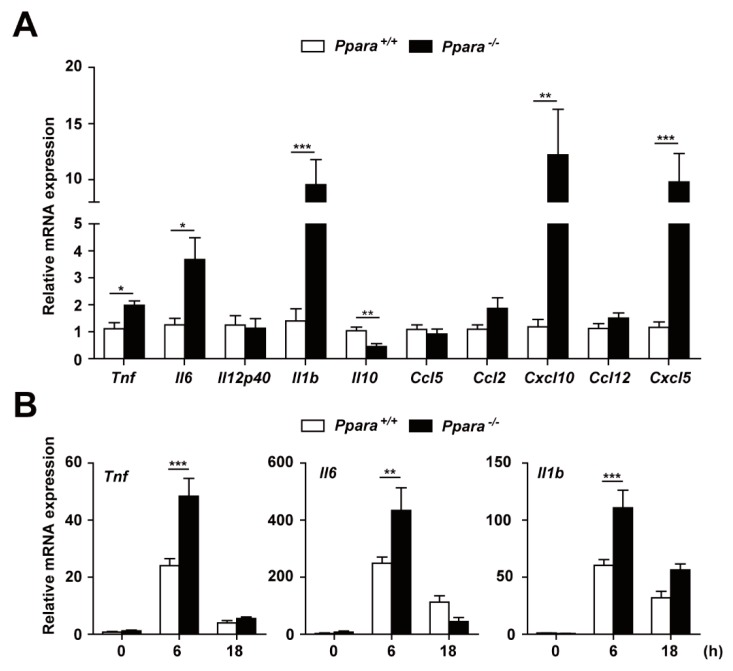
PPARα protects against excessive inflammation in Mabc-infected mice and macrophages. (**A**) Proinflammatory cytokine and chemokine mRNA levels in lung tissue were determined by qRT-PCR, as in Figure 1A. (**B**,**C**) *Ppara*^+/+^ and *Ppara*^-/-^ BMDMs were infected with Mabc (MOI of 5) for the indicated times, and the levels of proinflammatory cytokines were measured by qRT-PCR (**B**) or ELISA (**C**). (**D**,**E**) *Ppara*^+/+^ and *Ppara*^-/-^ BMDMs were infected with Mabc (MOI of 1, 3, or 5), and the levels of proinflammatory cytokines were determined by qRT-PCR at 6 h (**E**) or ELISA at 18 h (**D**). Data are means ± SD of three independent experiments. * *p* < 0.05, ** *p* < 0.01, and *** *p* < 0.001. UI, uninfected.

**Figure 3 cells-09-00648-f003:**
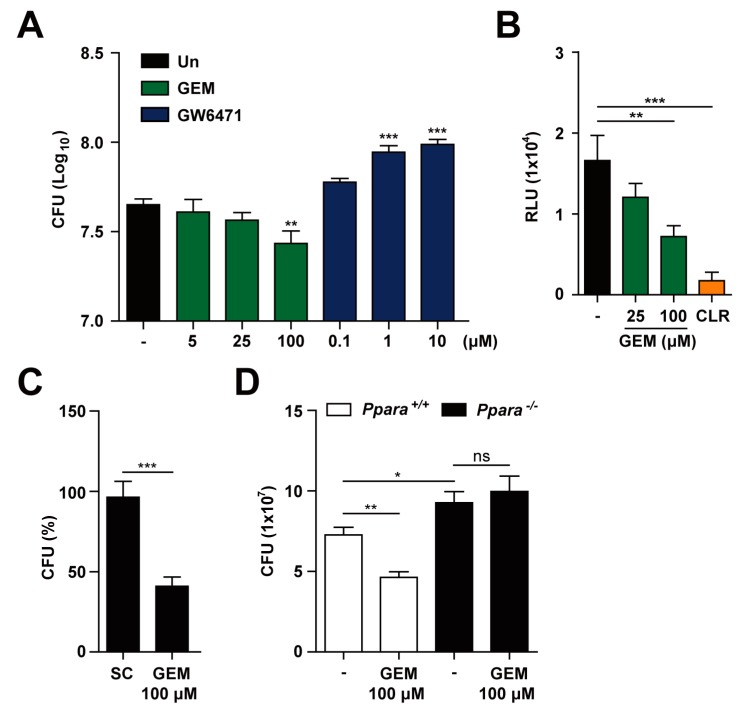
GEM promotes the antimicrobial response during Mabc infection in a PPARα-dependent manner. (**A**) BMDMs were infected with Mabc (MOI of 5) and treated with gemfibrozil (GEM; 5, 25, or 100 μM) or GW6471 (0.1, 1, or 10 μM) for 3 dpi. Intracellular survival of Mabc was determined by CFU assay. (**B**) BMDMs were infected with Mab*c*-LuxG13 (MOI of 5) and treated with GEM (25 or 100 μM) or clarithromycin (CLR; 5 μg/mL) for 3 dpi. Intracellular survival of Mabc was determined by bioluminescence analysis. (**C**) Human MDMs were infected with Mabc (MOI of 5) and stimulated with GEM (100 μM) for 3 dpi. Intracellular survival of Mabc was determined. (**D**) *Ppara^+/+^* and *Ppara^-/-^* BMDMs were infected with Mabc (MOI of 5) and treated with GEM (100 μM) for 3 dpi. Intracellular survival of Mabc was measured by CFU assay. Data are means ± SD of three independent experiments. * *p* < 0.05, ** *p* < 0.01, and *** *p* < 0.001. Un, untreated; SC, solvent control; ns, not significant.

**Figure 4 cells-09-00648-f004:**
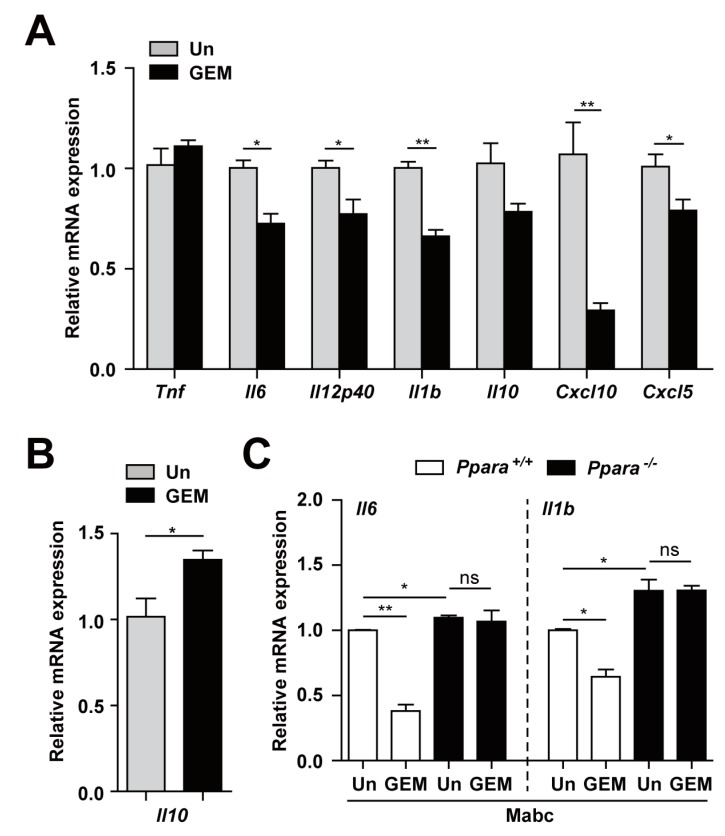
PPARα activation by GEM regulates the Mabc-mediated expression of inflammatory cytokines in a PPARα-dependent manner. (**A**,**B**) BMDMs were infected with Mabc (MOI of 5) and treated with gemfibrozil (GEM; 100 μM) for 6 (**B**) or 18 h (**A**). The mRNA levels of proinflammatory cytokines and chemokines were determined by qRT-PCR. (**C**) *Ppara^+/+^* and *Ppara^-/-^* BMDMs were infected with Mabc (MOI of 5) and treated with GEM (100 μM) for 18 h. The mRNA levels of *Il6* and *Il1b* were measured by qRT-PCR. Data are means ±SD of three independent experiments. * *p* < 0.05 and ** *p* < 0.01. Un, untreated; ns, not significant.

**Figure 5 cells-09-00648-f005:**
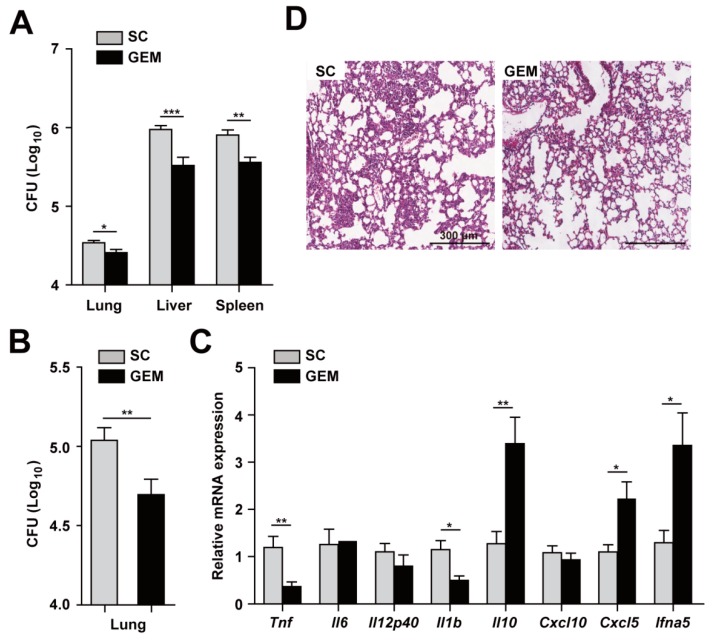
PPARα activation increases host protective responses and modulates inflammation during Mabc infection. (**A**) Mice (n = 5) were infected with Mabc (iv, 1 × 10^7^ CFU) and treated with the solvent control (SC; 0.1% methylcellulose) or gemfibrozil (GEM; 15 mg/kg/day, dissolved in 0.1% methylcellulose). The bacterial loads in the lung, liver, and spleen were determined at 7 dpi. (**B**) Mice were infected with Mabc (in, 3 × 10^6^ CFU) and treated with SC (0.1% methylcellulose) or GEM (15 mg/kg/day, in 0.1% methylcellulose), and the bacterial load in the lung was determined at 7 dpi. (**C**) Proinflammatory cytokine and chemokine mRNA levels in lung tissues determined by qRT-PCR, as in 5A. (**D**) Representative H&E-stained images of lung tissues. Scale bars, 300 μm. Data are means ±SD of three independent experiments. Images are representative of three independent experiments (**C**). * *p* < 0.05, ** *p* < 0.01, and *** *p* < 0.001. SC, solvent control.

**Figure 6 cells-09-00648-f006:**
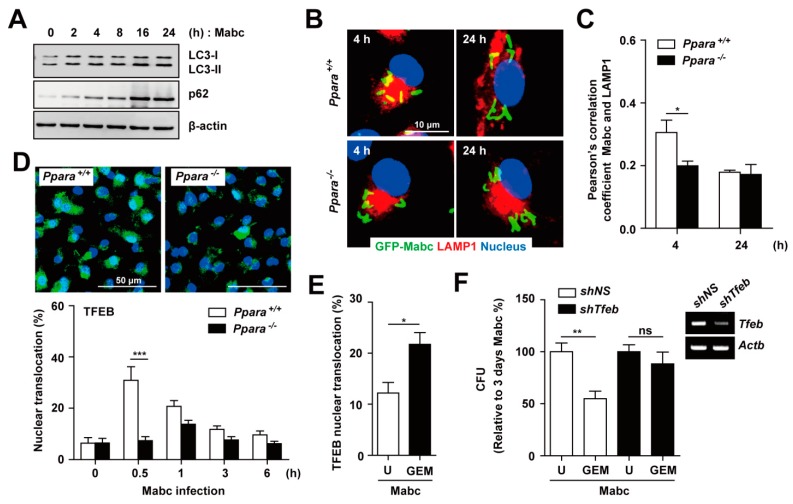
PPARα activation by GEM enhances TFEB-mediated antimicrobial activity. (**A**) BMDMs were infected with Mabc (MOI of 5) for the indicated times, and the p62, LC3, and β-actin protein levels were evaluated by immunoblotting (**B**) BMDMs were infected with GFP-Mabc (MOI of 5) for 4 or 24 h. Alexa-594-conjuated LAMP1 (red) and DAPI (nuclei; blue) were visualized by confocal microscopy. Scale bars, 10 μm. (**C**) GFP-Mabc and LAMP1 colocalization was assessed by calculating Pearson’s correlation coefficient. (**D**) *Ppara*^+/+^ and *Ppara*^-/-^ BMDMs were infected with Mabc (MOI of 5) for the indicated times, and nuclear translocation of TFEB was assessed by confocal microscopy. (Top) Representative images. Scale bar, 50 μm. (Bottom) Quantitative analysis of TFEB nuclear translocation. (**E**) BMDMs were infected with Mabc (MOI of 5) for 3 h and treated with gemfibrozil (GEM; 100 μM). Nuclear translocation of TFEB was assessed by confocal microscopy and calculated quantitative analysis. (**F**) BMDMs were transduced with lentivirus expressing a nonspecific shRNA (sh*NS*) or shRNA targeting *Tfeb* (sh*Tfeb*) using polybrene (8 mg/mL). After 36 h, BMDMs were infected with Mabc (MOI of 5) and treated with GEM (100 μM). (Right) Semiquantitative PCR analysis was performed to assess the transduction efficiency. (Left) Bacterial load was determined at 3 days. Data are means ± SD of three independent experiments. Images are representative of three independent experiments (**A**,**B**,**D**). * *p* < 0.05, ** *p* < 0.01, and *** *p* < 0.001. U, untreated; ns, not significant.

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
