# Peer review of "The Peroxisome Proliferator-Activated Receptor α- Agonist Gemfibrozil Promotes Defense Against Mycobacterium abscessus Infections"

_cells, 2020, doi:10.3390/cells9030648_

Round 1
Reviewer 1 Report
The study shows that host defense in mice may be enhanced - also in case of NTM infections -with the peroxisome proliferator-activated receptor PPAR(-alpha) receptor agonist gemfibrozil.
· The kind of stimuli- or tissue-specific activation of nuclear receptors determines signaling outcome in tissues. A major concern for comparing findings from different studies is the type of the chosen cell type, tissue or animal model used in the studies!
Thus, in the present study the experiments should be performed in one animal model. Particularly, data on the comparison of clarithromycin with a drug exerting probably more effects on host defense, namely gemfibrozil, should not be given in a Zebrafish model when all the data on mechanisms of action concerning gemfibrozil have been generated in a mouse model.
Following these considerations it would be also appropriate to test the drug combination clarithromycin and gemfibrozil in mice to show, whether additional activation of host defense by gemfibrozil may exert improved outcome in mice.
Also clarithromycin should be described as drug with pleiotropic activity profile.
· The transcription factor EB levels can be increased by retinoic acid or gemfibrozil. If the study is lastly geared towards the introduction of an 'old' drug in a novel clinical setting (drug repurposing), drugs with less side effects should be tested, additionally, probably also PPARalpha/gamma agonists.
· It should be stated that PPARalpha (and PPARgamma) expression is generally necessary for modulating host defense to microbial infections (introduction).
· Dynamics and functions of PPAR-α are modulated by the types of ligands that bind to the nuclear receptor. Therefore, it should be clearly worked out that gemfibrozil's anti-inflammatory activity may to be also mediated independently of its PPAR-α activity and that there is a direct anitmicrobial activity.
· Data on ‘pure’ antimicrobial activity of gemfibrozil should be presented, additionally.
· Data on drug concentrations of gemfibrozil used in mice should be discussed in comparison to those achieved in humans for addressing important translational aspects.
Author Response
Response to Reviewer 1 Comments
Comments and Suggestions for Authors
1. The study shows that host defense in mice may be enhanced - also in case of NTM infections -with the peroxisome proliferator-activated receptor PPAR(-alpha) receptor agonist gemfibrozil. The kind of stimuli- or tissue-specific activation of nuclear receptors determines signaling outcome in tissues. A major concern for comparing findings from different studies is the type of the chosen cell type, tissue or animal model used in the studies! Thus, in the present study the experiments should be performed in one animal model. Particularly, data on the comparison of clarithromycin with a drug exerting probably more effects on host defense, namely gemfibrozil, should not be given in a Zebrafish model when all the data on mechanisms of action concerning gemfibrozil have been generated in a mouse model.
- Thanks for your valuable comments. As you recommended, we deleted the data of zebrafishes in our revised work.
2. Following these considerations it would be also appropriate to test the drug combination clarithromycin and gemfibrozil in mice to show, whether additional activation of host defense by gemfibrozil may exert improved outcome in mice.
- We investigated any additional/synergistic effect on antimicrobial responses by the combined treatment of clarithromycin (CLR) and gemfibrozil (GEM) upon Mabc infection in vivo. To examine this, we treated mice with single drug and a combination for 5 days and performed the cfu assay. In vivo bacterial loads in the mice after treatment with drug combinations (CLR and GEM) were comparable with those treated with GEM alone. These data suggest that the combined treatment of mice with GEM and CLR has no additional antimicrobial effects upon in vivo Mabc infection. However, we think that further experiments for determination of dose/duration will be needed. Therefore we show the data as Figure 1 for the review purpose only.
3. Also clarithromycin should be described as drug with pleiotropic activity profile.
- The description has been included in line 292-293, as follows:
Results; Page 9 line 278-279)
However, it is noted that clarithromycin is a drug with pleiotropic activity (Kiesewetter B et al., (2018) Case Rep Oncol. Apr 11;11(1):239-245).
4. The transcription factor EB levels can be increased by retinoic acid or gemfibrozil. If the study is lastly geared towards the introduction of an 'old' drug in a novel clinical setting (drug repurposing), drugs with less side effects should be tested, additionally, probably also PPARalpha/gamma agonists.
- Since we found a little protective effect by GW7647 in our study (see Fig. S3), we examined the effect of GW7647 on the nuclear translocation of TFEB in macrophages after Mabc infection. As shown in Fig. S4, we found that there was no significant effect of GW7647 in the enhancement of nuclear translocation of TFEB in Mabc-infected BMDMs. These data suggest that GEM has a superior effect upon TFEB activation, when compared with other PPARα agonists.
These data have been included in Results (p16), as follows:
Results; Page 12, lines 361-368)
We next examined whether PPARα activation by GEM induces nuclear translocation of TFEB in BMDMs. GEM treatment of BMDMs led to a significant increase in TFEB nuclear translocation in Mabc-infected BMDMs (Figure 6E). We further examined whether other PPARα agonist regulated nuclear translocation of TFEB. As shown in Supplementary Figure S4, treatment of BMDMs with GW7647 did not significantly upregulate the nuclear translocation of TFEB in BMDMs during Mabc infection, when compared with untreated controls. Furthermore, the GEM-mediated antimicrobial response in BMDMs was significantly attenuated by knockdown of Tfeb using shRNA targeting Tfeb, compared with a nonspecific shRNA (Figure 6F).
5. It should be stated that PPARalpha (and PPARgamma) expression is generally necessary for modulating host defense to microbial infections (introduction).
- The description has been included in Introduction (p2), as follows:
Introduction; Page 2, lines 60-61)
Indeed, PPARα as well as PPARγ play crucial role in modulation of host defense against a variety of microbial infection [15-18].
6. Dynamics and functions of PPAR-α are modulated by the types of ligands that bind to the nuclear receptor. Therefore, it should be clearly worked out that gemfibrozil's anti-inflammatory activity may to be also mediated independently of its PPAR-α activity and that there is a direct anitmicrobial activity.
- Thanks for your valuable comments. To examine whether GEM’s anti-inflammatory activity is dependent on Ppara activity, Ppara+/+ and Ppara-/- BMDMs were infected with Mabc (MOI of 5) for 2 h and treated with gemfibrozil (GEM, 100 μM) for 18 h. The mRNA levels of Il6 and Il1b were determined by qRT-PCR analysis. As shown in Fig. 4C, we found that the inhibitory effects of GEM on Mabc-mediated Il6 and Ilb expression in Ppara+/+ BMDMs were significantly abrogated in Ppara-/- BMDMs. In addition, we found that GEM has no pure and direct antimicrobial activity against Mabc in vitro (see below). Therefore, in our revised work, we examined whether GEM-mediated antimicrobial effects depend on Ppara expression by using intracellular survival assay of Mabc in Ppara+/+ and Ppara-/- BMDMs in the presence or absence of GEM. Similar to the findings of GEM-mediated inflammatory responses, we found that GEM-mediated antimicrobial responses in Ppara+/+ BMDMs were markedly attenuated in Ppara-/- BMDMs (Fig. 3D). These data suggest that GEM activates the antimicrobial effects against Mabc infection through PPAR-α. These comments have been included in Results and Discussion, as follows:
Results; Page 9, lines 281-283)
Moreover, GEM-mediated antimicrobial activity was mediated through PPARα, since the GEM-induced suppressive effect on intracellular growth of Mabc in Ppara+/+ BMDMs was significantly abrogated in Ppara-/- BMDMs (Figure 3D).
Discussion; Page 14, lines 459-461)
However, GEM treatment showed a significant antimicrobial effect against Mabc infection in vitro and in vivo, depending on PPARα.
7. Data on ‘pure’ antimicrobial activity of gemfibrozil should be presented, additionally.
- In our revised work, we examined whether GEM had a direct and pure antimicrobial activity against Mabc. To examine this, we determined minimum inhibitory concentration (MIC) using resazurin microtiter assay (REMA). As shown in Fig. S2, we found that GEM has no direct antimicrobial response against Mabc in vitro. We apologize for our mistake of the wrong statement in our original manuscript, in Discussion, that GEM exerted a direct antimycobacterial effect. In our revised work, we have included the REMA results, and confirmed that GEM has no direct anti-Mabc effect in vitro.
New data and comments have been included in Materials and Methods and Results, as follows:
Materials and Methods; Page 5, lines 198-207)
2.13. Minimum inhibitory concentration (MIC) determination using resazurin microtiter assay (REMA)
The MICs of the compounds were determined as described previously (Kim TS et al., (2017) Antimicrob Agents Chemother. Aug 24;61(9)). Briefly, 100 μl of cation-adjusted Mueller–Hinton (CAMH) and 7H9 broth supplemented with 10% ADC was added to every well of a 96-well microtiter plate. A two-fold serial dilutions were prepared in 96-well clear microplates to obtain concentration ranges from 200 μM to 0.39 nM. Log-phase Mabc strain CIP 104536T cultures were added to each wells to an OD600 of 0.05 and the plates were incubated at 37°C for 3 days prior to addition of resazurin (0.025% [wt/vol]). After overnight incubation, fluorescence was measured by excitation at 530 nm and emission at 590 nm using a SpectraMax® M3 Multi-Mode Microplate Reader (Molecular Devices, CA). The MIC was calculated using Prism 6 software (GraphPad Software, Inc., La Jolla, CA).
Results; Page 9, lines 273-274)
Although PPARα agonist GEM [29] did not exhibit any direct antimicrobial effect against Mabc (Supplementary Figure S2),
8. Data on drug concentrations of gemfibrozil used in mice should be discussed in comparison to those achieved in humans for addressing important translational aspects.
- We treated mice with gemfibrozil at a dose of 15 mg/kg of body weight/day via gavage as previously described (Jana M et al., (2012) J Biol Chem. Oct 5;287(41):34134-48). In adult humans, gemfibrozil is prescribed at a dose of 600–1200 mg/day, therefore dose used in our study (15 mg/kg) is human equivalent dose to mice. (Jana M et al., (2012) J Biol Chem. Oct 5;287(41):34134-48).
Reviewer 2 Report
In the current paper, Kim et al. propose an interesting study about the promising manipulation of PPARα activation as a therapeutic strategy for NTM disease.
Nontuberculous mycobacteria (NTM) and in particular Mabc can be considered the cause of serious pulmonary infections in both immunocompetent and immunocompromised individuals and it is unfortunately associated with a high rate of treatment failure.
With the purpose to understand the host-pathogen interactions and consistent with recent studies about the involvement of multiple pathways/factors in the pathogenesis of Mabc infection and host protective immunity, in this study the authors investigate the role of PPARα activation on the antimicrobial response to Mabc infection In vitro and in vivo. They follow a valid experimental design to give strong evidence of the involvement of PPARα activation in the antimicrobial responses during Mabc infection.
In my opinion the study is well conducted but I would strongly recommend to improve the introduction section and to more extensively comment the presented results.
Author Response
Response to Reviewer 2 Comments
Comments and Suggestions for Authors
In the current paper, Kim et al. propose an interesting study about the promising manipulation of PPARα activation as a therapeutic strategy for NTM disease.
Nontuberculous mycobacteria (NTM) and in particular Mabc can be considered the cause of serious pulmonary infections in both immunocompetent and immunocompromised individuals and it is unfortunately associated with a high rate of treatment failure.
With the purpose to understand the host-pathogen interactions and consistent with recent studies about the involvement of multiple pathways/factors in the pathogenesis of Mabc infection and host protective immunity, in this study the authors investigate the role of PPARα activation on the antimicrobial response to Mabc infection In vitro and in vivo. They follow a valid experimental design to give strong evidence of the involvement of PPARα activation in the antimicrobial responses during Mabc infection.
In my opinion the study is well conducted but I would strongly recommend to improve the introduction section and to more extensively comment the presented results.
- Thanks very much for your kind comments. We have included more extensive comment in Introduction and Results, in our revised work.
Reviewer 3 Report
In this work, the authors investigated the possible role of PPARα in protecting infections coursed by NTM. The experiments were well designed and conducted in a good standard manner. The results are convincing and the manuscript is well prepared. The English is good.
Although the mice are supplied as syngeneic animals, it will strengthen the manuscript if the authors can show, at least in a couple of representative mice, that the PPARα protein is absent from the PPARα-negative mice whereas its presence is detectable in the PPARα-positive mice in those tested mouse organs and tissues.
Author Response
Response to Reviewer 3 Comments
Comments and Suggestions for Authors
In this work, the authors investigated the possible role of PPARα in protecting infections coursed by NTM. The experiments were well designed and conducted in a good standard manner. The results are convincing and the manuscript is well prepared. The English is good.
Although the mice are supplied as syngeneic animals, it will strengthen the manuscript if the authors can show, at least in a couple of representative mice, that the PPARα protein is absent from the PPARα-negative mice whereas its presence is detectable in the PPARα-positive mice in those tested mouse organs and tissues.
- Thanks very much for your kind comments. Ppara-/- mice were kindly provided by Dr Gonzalez FJ. Ppara-/- mice were established by targeted disruption of 83 bp coding sequence of exon 8, which encodes the ligand binding domain of α isoform of the peroxisome proliferator-activated receptor gene, in mice (Lee SS et al., (1995) Mol Cell Biol. Jun;15(6):3012-22). Since we have no available anti-PPARα Ab to detect mutated/deleted form of PPARα in the lab, we designed the qRT-PCR primers and confirmed that mPpara expression was markedly downregulated or undetectable in various organs/tissues (in bone marrow, lung, liver, and spleen) in Ppara-/- mice. In addition, we have provided the genotyping data of the mice in Figure 2 for the review purpose only.